# The Evaluation of Social Farming through Social Return on Investment: A Review

**Francesco Basset** [1,2]

1 Department of Economics Engineering Society and Enterprise, University of Tuscia**,** Viterbo 01100, Italy; francesco.basset@unitus.it
2 Council for Agricultural Research and Analysis of Agricultural Economics—Policy and Bioeconomy (CREA-PB), Rome 00184, Italy

**Abstract:** In recent years, there has been a need for a shared methodology for evaluating social farming (SF) practices to verify not only their effectiveness but also their social and economic sustainability. The evaluation of SF has been highlighted using the methodology of the social return on investment (SROI) due to the potential of such approach regarding the quantification of social impact. The main purpose of this study is to provide an overview, through a systematic review, of the application of SROI to SF experiences to check the results comparability, both in terms of outcomes standardization and comparisons between SROI ratios. The results first show some similarities on the construction of outcomes that allow for the initial comparability of the results. Secondly, all the indicators calculated in the articles report a social return value of social farming projects that varies approximately from EUR 2 to EUR 3 per euro invested. Critical issues remain regarding the application of this methodology to SF practices, regarding the number of the applications of SROI to SF, the process of stakeholder engagement and the construction of outcome. There is a need for more studies that apply SROI to SF experiences in order to standardize the process of analysis.

**Keywords:** social arming; uantitative valuation; ocial return on investment

## 1. Introduction

Social farming is a branch of multifunctionality agriculture that produces numerous social, environmental and cultural benefits [1]. The purpose of this practice is to provide social, social-health, educational and re-employment services (Law 141/2015) for individuals from disadvantaged categories (Law 381/91). Social inclusiveness, gender equality and responsible production, commonly in conjunction with sustainable farming [2,3], biological agriculture and re-use of abandoned farmlands [4] with the One Health approach [3] are distinctive characteristics of social farming. In this context, the objectives of social farming practices are fully in line with the 2030 sustainable developmental goals [5,6], particularly SDGs 5, 8, 10 and 12 [7], aiming at the creation of added social and environmental values in the entire production system. In particular, the aforementioned characteristic of social farming directly to the achievement of the targets 8.5 "by 2030, achieve full and productive employment and decent work for all women and men, including for young people and persons with disabilities, and equal pay for work of equal value", 8.6 "by 2020, substantially reduce the proportion of youth not in employment, education or training", 10.2 "by 2030, empower and promote the social, economic and political inclusion of all, irrespective of age, sex, disability, race, ethnicity, origin, religion or economic or other status", 10.3 "ensure equal opportunity and reduce inequalities of outcome, including by eliminating discriminatory laws, policies and practices and promoting appropriate legislation, policies and action in this regard".

Interest in the multifunctional role of agriculture and its associated social and environmental benefits with it is steadily increasing [8–10], even though the number of subjects and institutional actors operational in social farming, particularly in relation to welfare [11], continue to remain limited [12]. In fact, social farming also contributes to the identification of new welfare scenarios by promoting innovative paths of co-design and co-implementation of services, through the active involvement of multiple actors [13]. Indeed, the need to innovate the current system by focusing on a renewed collaboration between the public and private sectors, the creation of networks that involve even new actors and the identification of shared solutions to complex problems has been expressed from many fields [14]. Nevertheless, although many experiences (including that of social farming) extensively demonstrate the soundness of such approach, policies seem to be evolving extremely slowly, being bound to logics of cost containment rather than improved outcomes. Studies highlighting the "cost-effectiveness" of some innovative approaches, of the effects on individuals, businesses, and the environment, can, therefore, be extremely useful in decision support.

The importance of evaluating these practices does not lie on the simple measurement of monetary value—closely linked to the capitalist worldview [15]—but on the necessity to raise the value of both social and environmental overall impacts that these phenomena generate with a methodological approach using the same currency of the economic revenues. This methodological approach has been recently developed is social sciences with the social return on investment (SROI) as a quantitative assessment tool [16,17]. SROI has also been already used [18–20] for evaluating the social return of multifunctional agriculture even though its application on social farming has not been made yet the subject of a comprehensive analysis, which would evaluate its actual effectiveness, replicability and level of standardization in its implementation and results, limiting its replicability and standardization in this specific field of interest. However, the applications of SROI of social farming has not been made the subject of a comprehensive view in its implementation and results, limiting its replicability and standardization.

The objective of this systematic review is to provide an overview of a specific methodology for the evaluation of social farming [4,18,20]: the SROI. This methodology has been chosen because it is the only model, among those used in social farming, capable of giving the quantitative proxies of the impacts, including monetary terms. There is a strong debate in the literature [21,22] about the potential of this method in the economic quantification of social benefits, in the construction of the outcomes [23] and in the financial proxies used to calculate the SROI. For these reasons, the main purpose here is to increase the knowledge of the application of SROI to the phenomenon of social farming to check the comparability in the results, both in terms of outcomes standardization and comparisons between SROI ratios.

## 2. Theoretical Background

### 2.1. The Different Methodologies to Evaluate Social Impacts

Cost–benefit analysis is the most used approach to estimate the strengths and weaknesses of economic activities. However, some critical issues have been raised on the effectiveness of cost–benefit analysis in identifying and measuring hard-to-monetize benefits, such as social benefits [24–27].

Several authors [28,29] have highlighted the need to identify a more complex evaluation methodology for social impact/benefits, and different methodological approaches have been proposed to identify and evaluate the social impact generated by economic activities [15,28,30–33], even though they are primarily centered on qualitative rather than quantitative monetary assessment [11].

Theory of change [33], which is an analytical tool used to identify and evaluate the entire history of change from the moment it begins to occur to the moment it ends, was proposed to evaluate social impact. Social impact assessment [31] is a method used to

identify the direct and indirect social impacts, but also to evaluate development projects in agricultural disputes [30]. Moreover, some authors has been using SWOT analysis [32] and social enterprise impact evaluation [15] to assess the impact generated by social cooperatives. These studies highlight the importance of identifying and evaluating the social impacts generated by the multifunctional agriculture practices deployed by social enterprises [15], but they focus less on quantifying the social benefits of social farming. Finally, SROI has been used in several studies [16,17,34–36] to quantify social benefits in monetary terms, especially in the mental health field.

### 2.2. The Evaluation of Social Farming

Social farming has been evaluated extensively from a qualitative perspective [4,37–41] but less from the quantitative angle. Most of the studies focus on describing social farming experiences and their effects on specific disadvantaged groups, with little attention, if any, on the quantification and the economic measurement of the effects/benefits produced.

In this context, the quantification of the social benefits of social farming has been highlighted in recent years by studies using SROI as a methodology [21,22,42], which has been developed and widely used internationally to evaluate projects and organizations in the field of public health [16]. The ability of this methodology [22] to quantify social impact makes it possible to extend this approach to different fields, such as social farming. In recent years, in fact, this approach has also found applications in social farming projects [18–20,42,43].

### 2.3. Social Return on Investment (SROI)

SROI [44] is a methodology that aims to quantify a broader concept of "value" than that expressed in economic or financial terms alone. Using outcomes, indicators and financial proxies to measure social and environmental benefits, allows the social and environmental components to be expressed in monetary terms. This approach, integrated with cost–benefit analysis, is able to value the reduction in social inequalities, environmental degradation and the improvement in the well-being of individuals resulting from a given investment.

There are two types of SROI analysis [44]:

1. Evaluative, conducted ex-post and based on actual outcomes achieved;
2. Predictive, aimed at predicting the social value that will be created if activities achieve the expected outcomes.

This methodology of analysis involves six steps for its successful implementation:

1. Defining the scope of analysis and identifying the stakeholders;
2. Mapping the outcomes;
3. Quantifying outcomes and assigning a value;
4. Defining the impact;
5. Calculating the SROI;
6. Returning, using, integrating.

The main feature of the SROI relates to the evaluation process used to measure social and environmental benefits (impacts), based on a few characterizing points:

(a) A comprehensive identification, and subsequent involvement, of stakeholders (direct and indirect) that are affected by the implementation of social farming projects;

(b) The creation of an impact map that identifies the effects (outcomes) that projects generate on identified stakeholders;

(c) The development of calculation tools (proxies) for quantifying, in monetary terms, these effects, which, in relation to the stakeholders, invest in the economic, social and environmental spheres.

### 3. Materials and Methods

The PRISMA approach was used [45] to describe the results obtained in this systematic review on the use of SROI to evaluate projects in social farming. The review choice relies on this method, as a systematic review characterized by transparent paper selection process reduces the effects of researcher bias and improves the rigor and completeness of the analysis [46].

#### 3.1. Research Strategy

Initially, a keyword search was conducted on "Scopus". The search terms initially used were "Social Return on Investment" or "SROI" and "Social Agriculture" or "Social Farming" or "Green Care". Later the search was expanded by the terms "evaluation" and "social impacts"; however, most of the papers mined with this approach dealt with "green care" and other topics that were not in the scope of this study.

Therefore, the search fields were limited to the more focused terms "SROI" or "Social Return on Investment" combined with "social farming" or "social agriculture". Moreover, the same search was performed on Google Scholar to identify papers potentially dealing with social farming. After an initial analysis of the papers found, a search was finally conducted by author, and the bibliography of each paper was analyzed to ensure that every study on the applicability of SROI to social farming was included.

For both the literature sources identified through Scopus and those identified on Google Scholar, a time order from 1996 to 2022 was included. This time frame was chosen because the first recorded SROI report was published in 1996 [16].

The analysis of the papers was conducted manually, first through an analysis of the abstracts and then by reading the full text of the articles. The eligibility of articles for the study was determined by defining inclusion and exclusion criteria.

#### 3.2. Criteria for Inclusion and Exclusion

Articles that provided both a theoretical framework on the applicability of SROI to social farming projects and an application of SROI through a case study were considered. On the other hand, articles that addressed the topic of social farming evaluation from a qualitative perspective, or through other methodologies and/or applied SROI to other fields of agricultural multifunctionality were excluded. Finally, those articles for which the full text could not be accessed were also excluded.

#### 3.3. Quality Assessment

The evaluation of selected articles was performed by scoring specific quality assessments. Specifically, a score from 1 to 3 was assigned for the following criteria:

1.  Quality of analysis includes transparency about the rationale for using the SROI and its application;
2.  Transparency in the construction and quantification of outcomes;
3.  Sample size: how representative the study is of the overall phenomenon;
4.  Reflection on the results: in particular, whether a discussion of the strengths and limitations of this methodology was assessed.

For criteria assigned a value of 1 point, a rating of "low quality" was given. For a score of 2, the rating was "medium quality". Finally, for those criteria assigned a value of 3 points, the items were rated "high quality".

#### 3.4. Data Extraction and Summary

Descriptive analyses were first carried out on the selected articles. In particular, the year of publication, the country in which the article was written, the presence or absence of case studies within the articles, and the types of service users studied were analyzed.

Next, the issues of particular interest in achieving the set goals were analyzed. Then, where possible, the degree of stakeholder involvement and the types of stakeholders included in the analysis, the outcomes that were constructed for the SROI analysis, the SROI ratio obtained, the time horizon of the analysis, and the applicability of the SROI for social farming evaluation were analyzed. The last issue was analyzed by summarizing the strengths and weaknesses of SROI expressed in the identified articles.

The results obtained were used to map the applications of SROI to social farming.

## 4. Results and Discussion

### 4.1. The PRISMA Approach and the Choice of Quality Assessments

An initial analysis of article abstracts identified 57 scholarly articles on the application of SROI to the social farming phenomenon. Then, based on the eligibility criteria listed above, the sample was reduced to 10 articles. The remaining 47 articles were excluded from the study because after a thorough reading of the full texts, they were found not to be relevant to the object of study of this systematic review. In addition, one final paper was eliminated because the case study was a duplicate of an article already in the analysis. Of the nine selected articles, 45% came from the Scopus database, while the remaining 55% came from articles identified through the Google Scholar search engine. Of the latter, 95% were scholarly articles, while 5% were doctoral dissertations (Figure 1).

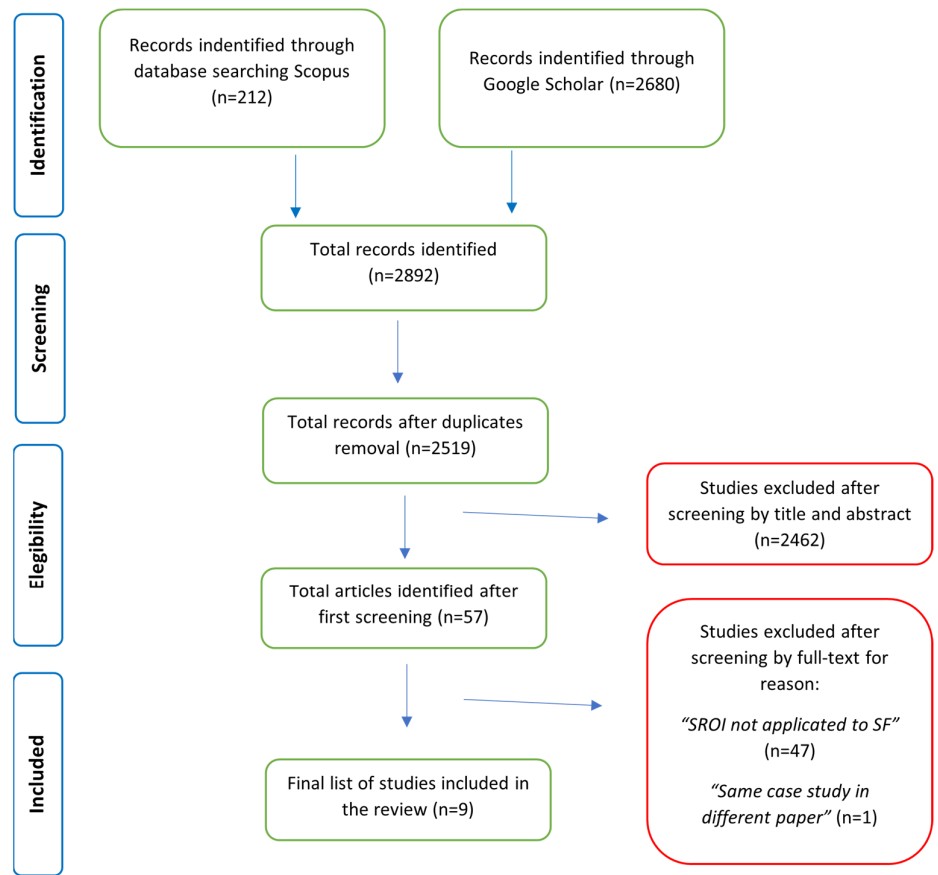

**Figure 1.** The PRISMA approach.

After the process of identifying and admitting articles, the articles were evaluated through quality assessments (Figure 2). The quality assessments were chosen and quantified according to the research objective of this systematic review.

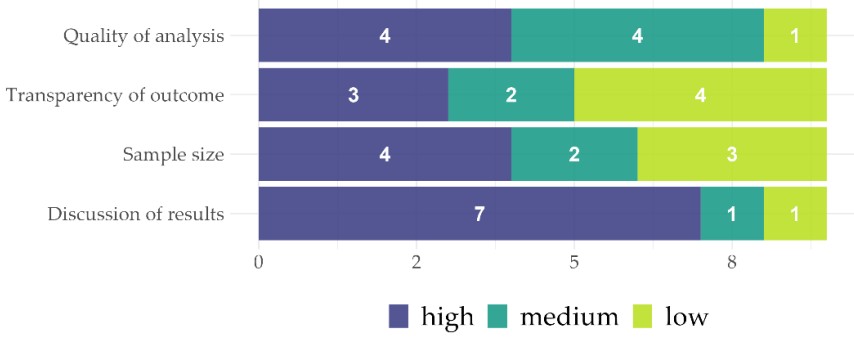

**Figure 2.** Quality assessment.

Regarding the criterion "Quality of Analysis", 45% of the articles comprehensively show the analysis design, the reasons why and how the SROI was applied. Therefore, these articles are found to have a "high" value, 45% a "medium" value, and the remaining 10% a low value. The articles evaluated according to the criterion "Transparency of outcomes" are 30% high-quality articles, as they have the description of the outcome map within them. The remaining 70% consisted of 23% medium-quality articles and 47% low-quality articles. Concerning the "Sample size" criterion, 45% of the articles were rated "high" in value, as the analysis sample was representative of the phenomenon under consideration, 33% were medium in value and the remaining 22% were low in value. Finally, the articles evaluated according to the criterion "reflection on the results" are 78% high-quality articles, as they presented in discussion the limitations and critical issues of this method. The remaining 22% are medium- and low-quality articles.

*4.2. Descriptive Analysis*

The descriptive analyses carried out on the selected articles regarded the year of publication, the geographical area of the studies, and the research methodology for applying SROI to the evaluation of social farming (case study or theoretical framework).

Regarding the year of publication, it can be observed (Figure 3) how the time range of publications is from 2013 to 2021. It can also be seen how the articles are distributed linearly. There is, in fact, no strong predominance of one year over another.

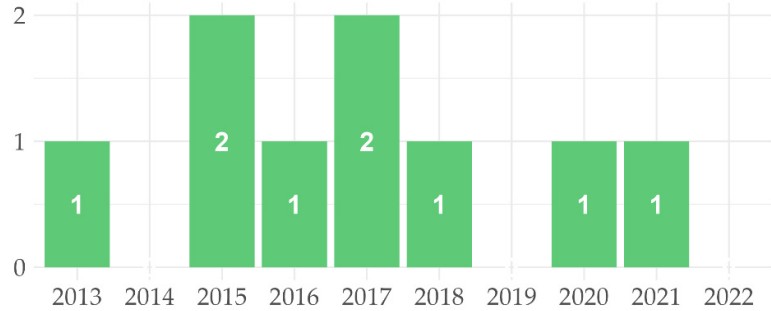

**Figure 3.** Year of publication.

Regarding the geographical distribution in which the studies were conducted, as shown in Figure 4, there is a predominance of studies in Spain (three studies). The remaining studies were conducted in the UK (two), Italy (one), Germany (one), Ireland (one) and Mexico (one). The predominance of studies in Spain can be explained not only by the characteristics of the area in which the phenomenon was analyzed but also by a large group of researchers interested in these issues.

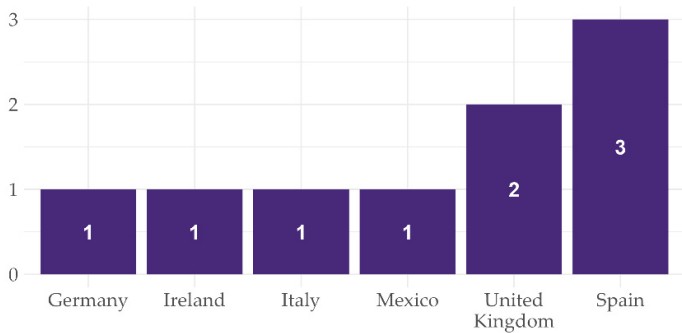

**Figure 4.** Distribution of studies.

Finally, Figure 5 shows the research methodology used. Almost all studies present an application of SROI to a specific case study (77 percent). The remaining 23% consisted of two studies that provided a theoretical framework on the potential and limitations of using SROI to evaluate social farming.

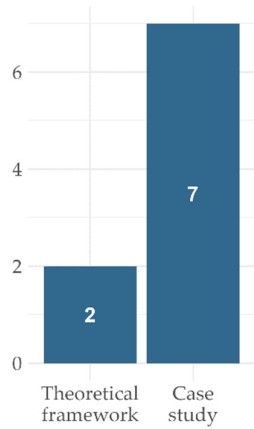

**Figure 5.** Research methodology.

*4.3. Research Theme*

4.3.1. Identification of Stakeholders

As a first point, we wanted to analyze, in the articles that provided an application of the SROI, which stakeholders were involved in the analysis and the relevance of the effects generated by the social farming on the different categories of stakeholders.

All studies analyzed have identified the following categories of stakeholders, i.e., the service users of the projects, the social farms or farms hosting the internship services, the local institutions and the community in which the projects were embedded.

Moreover, different categories of service users were considered in the studies identified and analyzed (Figure 6), i.e., people with disabilities, people at risk of social exclusion and people affected by addiction. This last category represented only 10% of the sample, with a predominance of the studies targeting people with disabilities and/or people at risk of social exclusion (90 percent).

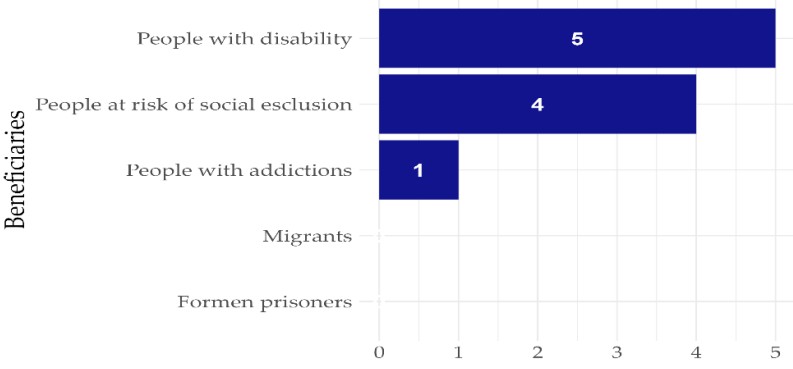

**Figure 6.** The service users of social farming projects.

Depending on the various purposes of the project, different entities related to specific services (addiction recovery centers, social recovery centers, group homes and associations) were included among the stakeholders.

In addition, 71% of the studies included the service users' family members as stakeholders in their analysis since many studies [4,19,20,43] have emphasized that, especially in cases of projects targeting individuals with disabilities, there are important spillovers to their family members as well. A few studies also included volunteers in the stakeholder-mapping [19,20,43] and one [18] included as a stakeholder the environment since it has already been considered as a subject with a legal personality [47].

### 4.3.2. Mapping Outcomes and Impacts

One of the most debated aspects on the use of SROI for social agriculture evaluation is the construction of outcomes [22,23]. For this reason, one of the main topics of this paper is to compare the outcomes constructed in the different articles analyzed, towards a standardization in the construction of outcomes at least for those stakeholders common to all studies. It has been possible only for the three studies [18,20,43], out of the nine selected, which presented a clear description of the construction procedure of financial outcomes and proxies through an impact map. Another two studies [4,48] presented the effects on stakeholders in the form of outcomes, but they did not present the procedure used for their quantification.

In the three studies where the outcomes and impacts map were analyzed, the outcomes and proxies were constructed through interviews or sources in the literature.

By analyzing the outcomes (Table 1) for different stakeholders, the study shows that for the category of users, the identified outcomes are "improved quality of life", "acquisition of job skills", and "reduced social isolation" in all studies. Additionally, additional outcomes such as "increased independence" are considered in some studies. There is also similarity in the construction of proxies for quantifying outcomes; for "acquisition of job skills" and "reduced social isolation", the proxies "remuneration from post-project contracts" and "lower cost for psychological sessions" are used.

**Table 1.** The map of outcomes in common among three studies [18,20,43].

| | Outcomes | Proxy | Basset e Giarè, 2021 | Leck, 2013 | Tulla et al., 2020 |
|---|---|---|---|---|---|
| **Beneficiaries** | | | | | |
| | improve quality of life | salary riceived for work performance | X | X | X |
| | acquired new skills | salary of new job after the project | X | X | X |

| | | | | | |
|---|---|---|---|---|---|
| | Improved mental health/reduce social isolation | fewer visits for psicological recovery | X | X | X |
| | improved social relationship | People who participate in social and/or cultural activities | | X | X |
| **Cooperatives** | | | | | |
| | development of cooperatives | lower cost of production | X | | |
| | | number of people in cooperative who feel satisfied about their job | | | X |
| **Host Farms** | | | | | |
| | increase of marketplace value | increase sales | X | X | |
| **Health Service** | | | | | |
| | lower hospital/caregiver costs | daily cost of hospitalization | X | X | X |
| **Families** | | | | | |
| | less time spent on care | lower cost for family therapy | | X | X |
| **Community** | | | | | |
| | decrease in unemployment rate for disabled people | lower cost of unemplyment benefits | | X | X |
| | lower cost of social security | daily cost for prisoners | X | | |
| | | value of social security payments | | | X |
| **Enviroment** | | | | | |
| | Improvement of the biodiversity of the territory around the project | Less dispersion of ecosystem value for recycling xenobionts | X | | |
| | | proxy identified but not measured | | | X |

Outcomes for local institutions and local communities depend greatly on the type of stakeholders analyzed. In fact, the mapping of outcomes in studies that dealt with different categories of subjects, made it possible to highlight and quantify with a unified methodological approach, different social impacts on different local communities, depending on the category of subjects under study. Therefore, for projects that have people with physical or mental disabilities as service users, measurable social benefit are observed through decreased costs for physical and psychological recovery and a decrease in the unemployment rate for this category [20,43].

For projects targeting individuals with addiction [18], social gain that can be measured through a decrease in the likelihood of these individuals committing crime (in turn measured through the cost of incarceration expenses).

Regarding health national services, a lower number of hospital admissions can be measured through the daily cost of hospitalization.

Comparisons can also be made for the outcomes constructed for cooperatives and farms. In fact, in the two out of three studies analyzed, the main outcome was

"Development of the cooperative" measured through the proxy "lower cost of production". Thus, an increase in revenues and sales due to lower product cost was found. Regarding farms, there is a concern around the increase in marketplace value, which is measured with the proxy "increase sales".

In studies that included family members of users, the best benefit identified was less time spent on care. In both studies this outcome was measured by the lower cost of family sessions for psychological support.

It was not possible, however, to find similarities in the construction of outcomes for project volunteers. In addition, particular differences were found in the construction of various outcomes, which are mostly to be attributed to differences in the identification of stakeholders or, as mentioned earlier, to differences in the projects analyzed.

### 4.3.3. SROI Ratio

Regarding the calculation of the SROI indicator, there is a differentiation among the selected items. A total of 57% apply SROI to a single project, repeated over time by aggregating the costs and benefits of different years into a single indicator. In contrast, 43% of the articles apply the SROI to multiple projects, managed by different social farms. These articles provide both an indicator with the aggregated costs and benefits for each project and the different indicators calculated for each project.

In each case, all the indicators calculated in the articles report a social return value of social farming projects that varies approximately from EUR 2 to EUR 3 per euro invested.

### 4.3.4. The "Time" Factor

Time is a factor to be considered in SROI analysis in two different aspects. The first concerns the boundaries of the analysis, i.e., whether it is used to evaluate a project or the performance of a company or institution. In the first case the boundary will be the duration of the project; in the second, a choice must be made about the years to be considered in the analysis [44].

All the papers analyzed in this review chose to analyze individual projects, aggregating, as seen, the results for multiple years or multiple social farms. This choice is supported in the literature [16,28] by the potential this methodology has demonstrated in evaluating individual projects.

The second aspect concerns the duration of outcomes, and consequently, the duration of the calculated social return value. This aspect is shown in 43% of the articles. Among them, 67% consider the drop-off (the loss of value over time) to be 0 for all outcomes. For these studies, the longer the duration of the identified outcomes, the greater the effect of that outcome on the stakeholder. In contrast, 37% of the articles calculate drop-off for only three outcomes out of the fifteen measured.

### *4.4. Discussion of Results*

This systematic literature review allowed us to map and to give comprehensive analysis of existing studies on the application of SROI to social farming practices, highlighting the effectiveness of SROI in quantifying social impacts. Indeed, all the studies analyzed have shown an economic return, ranging on average from two to three times the investments, characterized by a strong social component. Moreover, the overall view of the phenomenon made it possible to identify what areas still need to be explored and what interventions are needed to further extend and consolidate the implementation and evaluation of social farming experiences.

Identifying articles was the first challenge of this study, as there is no indexed database for the applications of SROI [16].

The fact that the first applications were made in the UK is no accident. In fact, SROI was designed and developed in the UK [49], and this finding is consistent with other studies on SROI in other areas [16,50]. The distribution over the years 2013–2021 is also

consistent with reviews in the literature on applications of this methodology [16,46,50,51]. Nowadays, the multifunctionality of agriculture is assuming an increasingly important role, especially in view of the new Agenda 2030 goals [6]; [40], and the interest in the role these practices play in creating social value is growing [2,39,52].

Regarding the geographical distribution of studies, as it has been seen, there is a strong predominance of Spain and the UK. For the UK this can be explained by what has just been said about the temporal distribution. For Spain on the other hand, the motivation lies in the presence of a growth interest in the social farming topic for rural development plans [2,53,54].

Similarities in the identification of key stakeholders (service users, cooperatives/farms, local institutions and the local community) were observed. In addition, national personal care services are included in all studies. The identification of institutions and health national servicers as key stakeholders confirms the importance of public–private sector collaboration to efficiently implement these types of services.

Different authors did not always include indirect stakeholders, such as the case of users' family members, in the analysis. Certainly, differences in the projects that were evaluated and differences in the territorial areas in which they are implemented also play a role in this context.

Analyses on the construction of outcomes showed that, at least on the studies carried out so far, it was possible to make an initial standardization regarding service users, regardless of their category. In addition, it was seen that even for the categories of local institutions and communities, an initial standardization can be obtained. It is necessary, however, to differentiate items by category of service users as different indirect effects on society were observed for different categories of service users. Clearly this also depends on the many differences among the social farming projects evaluated in the articles. The first standardization for key stakeholders is one of the main results of this work, as it allows the reduction in the analyzer bias by estimating the SROI ratio. In fact, the studies analyzed show that at least 70% of the result comes from quantifying the outcomes constructed for the key stakeholder categories.

The calculation of the SROI ratio shows how these projects are sustainable in all their dimensions; all studies show more than positive social returns. Comparisons on SROI ratios have shown the economic relevance of social, and to some extent, environmental, returns. In fact, it should be noted that this methodology allowed some studies [18,20] to break down the outcome obtained into the economic, social and environmental components, showing that more than one-third of the social return comes from measuring social outcomes. This, for the authors, is the greatest potential of applying SROI to this context. Indeed, the importance of quantifying social outcomes [15] for the evaluation of social farming should not lie in the classic view of business reporting. Rather, it lies in the possibility of valuing and making observable and comparable, social benefits on a par with economic ones. Only then can they be factors that countries can consider in policy choices. Moreover, in addition to making it possible to break down the result into the economic social and environmental components, SROI makes it possible to distribute those result among all identified stakeholders; the amount of indirect benefits accruing to society was shown, for example. In all studies, the breakdown of SROI ratios showed the same distributions of value generated. This is further evidence of the significance of the social and environmental benefits generated by social farming. The results on the SROI ratio highlight both which aspects of social farming need more intervention from institutions and the importance of the benefits that accrue to indirect stakeholders, such as society and the environment.

The analysis of the drop-off of social farming projects, which showed the absent of any drop-off in all projects considered in this review, emphasizes the importance of giving continuity to social farming experiences, going beyond individual projects in the direction of integrated national programs, enabling a more efficient implementation of these practices.

To this aim, it is surely needed to expand the current knowledge on social farming by increasing and strengthening the case studies and to further improve the methodological standardization, particularly with regard to the process of identifying stakeholders and their respective outcomes. This review is a first step in this direction; a larger sample of applications of SROI to social farming would allow a deeper evaluation of the methodological standardization already achieved for a few stakeholder categories and an extension of its application to other stakeholder categories, which have not been addressed so far.

## 5. Conclusions

A strong interest has emerged in the literature in the evaluation of social farming. Few studies, however, have proposed quantitative methodologies for this purpose. The SROI methodology has been found to be adequate for the evaluation of social farming projects in the studies in which it has been proposed, but there are still unexplored areas of research within this methodology, such as sensitivity analyses of estimates. Further studies on the applications of SROI to agriculture projects would allow for greater standardization of the processes of identifying stakeholders and constructing outcomes and proxies. All the studies analyzed had more than positive SROI ratios, demonstrating the sustainability of these practices.

In conclusion, there is a need for studies that apply impact assessment methodologies to the phenomenon of social farming. The aim is to find a standardized and shared methodology that can enhance social farming as a relevant practice in achieving the sustainability goals of Agenda 2030.

**Funding:** This research received no external funding.

**Informed Consent Statement:** Informed consent was obtained from all subjects involved in the study.

**Acknowledgments:** I sincerely thank my supervisors Francesca Giarè and Saverio Senni for the continuous suggestions and discussions we have had on this topic. In addition, their comments on an earlier version of this manuscript were crucial to its improvement.

**Conflicts of Interest:** The authors declare no conflict of interest.

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
