# Peer review of "The Evaluation of Social Farming through Social Return on Investment: A Review"

_sustainability, doi:10.3390/su15043854_

Round 1
Reviewer 1 Report
Dear Author,
The topic presented is very interesting and the manuscript is written in a clear and fluid way. I would like to congratulate you for the excellent work and for the way you have approached the study and broken it down to provide a comprehensive and in-depth overview.
The Introduction and Theoretical Background sections are well structured and written in a very fluid way which makes it an enjoyable read. It provides an adequate background where the author provides not only some concepts, but also provides a good overview of what has been published in the area of the study and is well referenced.
The Materials and Methods section is also very well written and clear. The Results and Discussion section is also fluidly written, with a well-founded Discussion from which several conclusions can be drawn.
The only limitation is the number of articles, but the universe of publications on the subject studied is reduced and I congratulate the author for having referred to this and the limitations resulting from it.
Below are some more comments that may contribute to improve the manuscript, the most relevant ones first, followed by minor corrections.
Major comments
1. The referencing in the text is different from that stipulated in the Journal Template. According to the instructions, "References should be numbered in order of appearance and indicated by a numeral or numerals in square brackets-e.g., [1] or [2,3], or [4-6]."
2. I would also ask the author to place the list of References according to the template and to make them uniform.
Minor comments
1. Before the title of the article, according to the template, it should be "Review" and not "Systematic Review".
2. Line numbering is missing, as defined in the Journal's template.
3. Page 1: “The objectives of social farming practices are fully in line with the Agenda 2030 Sustainable Developmental Goals (Marchis et.al, 2019), particularly SDGs 5, 8, 10 and 12”. Perhaps it would be clearer and more immediate for the reader if reference were made to what these SDGs 5, 8, 10 and 12 are.
4. At the top of page 2: “In addiction, quantifying the impacts in monetary terms could ensure greater spread of this practices”, isn't it "In addition" instead of "In addiction"?
5. Page 2: “This methodology is chosen because, being the only one, among those used in social farming, giving quantitative proxies of the impacts, even in monetary currency.” Perhaps it would be advisable for the author to rewrite this sentence because it is a bit confusing.
6. Page 3: “in this systematic review on the use of Social Return on Investment to evaluate projects in Social.” Maybe add "farming" at the end.
7. The title of section 4: "4-. Results and Discussion" has a hyphen between the 4 and the full stop.
8. Page 6: "Concerning the "Simple size" criterion, 45% of the articles were rated". I think it should be "Sample size" instead of "Simple size".
Author Response
I thank the reviewer for the valuable comments that helped us to significantly improve the paper.
Major comments
Point 1: The referencing in the text is different from that stipulated in the Journal Template. According to the instructions, "References should be numbered in order of appearance and indicated by a numeral or numerals in square brackets-e.g., [1] or [2,3], or [4-6]."
I thank the reviewer for this comment. I changed all references using the software Zotero, according to the Journal Template, and I numbered them in order of appearance, as kindly suggested.
Point 2: I would also ask the author to place the list of References according to the template and to make them uniform.
I revised all references using the software Zotero, according to the Journal Template.
Reviewer 2 Report
Dear authors,
thank You for presenting the review of a subject that shows scarce literature resources on social farming.
The paper may need some changes to the structure of the text:
for example :
in the section
2.1. The different methodologies to evaluate social impacts.
Cost-Benefits Analysis is the most used approach to monetize benefits from economic activities. However, some critical issues have been raised regarding the effectiveness of cost-ben-efits analysis in identifying and measuring hard-to-monetize benefits, such as social benefits (Cordes, 2017). Different methodological approaches have been
you should start from the broader perspective of:
Theory of Change (Galligani, 2019), is an analytical tool used to identify and evaluate the entire history of change from the moment it begins to occur to the moment it ends; the Social Impact Assessment (Becker, 2001), is used to identify the direct and indirect social impacts.
Take care in the other sections to the logical order as well.
Some technical propositions should be followed in the paper:
for example, in Figures 2 and 3 font paltino linotype should be used.
In references follow the technical proposition of MDPI ; bold, italics, points and doi numbers were not properly used.
Best regards,
the reviewer.
Author Response
I thank the reviewer for the important points risen, which helped me to substantially improve the clarity and effectiveness of the manuscript.
Main comments
Point 1: The paper may need some changes to the structure of the text: for example: in the section “2.1. The different methodologies to evaluate social impacts”. Cost-Benefits Analysis is the most used approach to monetize benefits from economic activities. However, some critical issues have been raised regarding the effectiveness of cost-benefits analysis in identifying and measuring hard-to-monetize benefits, such as social benefits (Cordes, 2017). Different methodological approaches have been.
You should start from the broader perspective of:
Theory of Change (Galligani, 2019) is an analytical tool used to identify and evaluate the entire history of change from the moment it begins to occur to the moment it ends; the Social Impact Assessment (Becker, 2001), is used to identify the direct and indirect social impacts.
Take care in the other sections to the logical order as well.
I revised the structure of the text in all manuscript, as kindly suggested. Specifically, the sections: 1 introduction, 2.1 The different methodologies to evaluate social impacts, and 4.4 discussion of results.
Point 2: Some technical propositions should be followed in the paper: for example, in Figures 2 and 3 font paltino linotype should be used.
I changed the font in palatino linotype in all Figures and Tables, as kindly suggested.
Point 3: In references follow the technical proposition of MDPI; bold, italics, points and doi numbers were not properly used.
I revised all references using the software Zotero, according to the Journal Template.
Reviewer 3 Report
A file is uploaded

Round 2
Reviewer 2 Report
Dear authors,
thank You for the changes made in the text according to all reviewers.
I noticed in the new document that Figure 7 is out of the frame of the paper - do check it technically before publishing.
All other technicall requirements are now met.
best regards and success with publishing,
the reviewer